# Functional recovery 2-years after hospitalization for COVID-19: Insights from the COREG-FR extension study

Marla Beauchamp[1]*, Christopher Farley[1], Renata Kirkwood[1], Aaron Jones[2], Terence Ho[3,4], MyLinh Duong[3,4], Parminder Raina[2], Jinhui Ma[2]

1 School of Rehabilitation Science, McMaster University, Hamilton, Ontario, Canada, 2 Department of Health Research Methods, Evidence, and Impact, McMaster University, Hamilton, Ontario, Canada, 3 Department of Medicine, McMaster University, Hamilton, Ontario, Canada, 4 Firestone Institute for Respiratory Health, St Joseph's Healthcare, Hamilton, Ontario, Canada

* beaucm1@mcmaster.ca

## Abstract

### Introduction

COVID-19 infection can lead to multi-organ dysfunction, which has been shown to contribute to the physical disability seen in people after hospital discharge. We aimed to understand the effects of hospitalization for COVID-19 on mobility, cognition, and daily activities over 24-months of follow up.

### Materials and methods

This was a 24-month extension of the COREG-FR prospective cohort study (NCT04602260). We enrolled consecutive adult patients (≥18 years) with lab confirmed SARS-Cov-2 infection who were admitted to five Ontario, Canada hospitals between August 21, 2020, and December 21, 2021. Patients were excluded if they resided in an institution (e.g., long term care facility), had severe premorbid physical function limitations (e.g., unable to stand independently) or had cognitive impairment which limited their ability to complete follow-up assessment. We assessed mobility and cognitive status using the Activity Measure for Post-Acute Care (AM-PAC) Basic Mobility Domain and Cognitive Domain, respectively. Deficits from premorbid status were determined using the minimal clinically important differences in mobility (≥ 3.3) and cognition (≥ 5.5). We also asked participants how much their COVID-19 recovery affected their daily activities within the preceding week with response options from 'not at all' to 'all the time'.

### Results

Among the 215 participants who participated 12-months after hospital discharge, 170 (79%) consented to the 24-month follow-up. The mean (standard deviation) age was 61.2 (12.7) years and 54% (n = 91) of participants who were male. Compared

**Data availability statement:** Data cannot be shared publicly because of privacy concerns (i.e., sensitive patient information). While our research ethics board approval precludes making the dataset publicly available, a minimal dataset can be made available upon request with approval from the Hamilton Integrated Research Ethics Board https://www.hireb.ca/hireb-contacts/.

**Funding:** The COREG Registry and COREG-FR are funded by a Canadian Institutes of Health Research grant (172754). This funding was awarded to MB. The funders had no role in study design, data collection and analysis, decision to publish, or preparation of this manuscript.

**Competing interests:** The authors have declared that no competing interests exist.

to pre-morbid function, mobility and cognitive deficits were present in 57% and 41% of participants, respectively. Furthermore, 59% of participants reported COVID-19 continued to impact their daily activities.

## Conclusion

At 24-months after hospitalization for COVID-19, many participants experience persistent mobility and cognitive deficits. Future work should aim to develop comprehensive rehabilitation strategies for those recovering from COVID-19 which target mobility and cognitive function.

## Introduction

Since the onset of the pandemic, over 777 million cumulative cases of COVID-19 have been reported globally [1]. Infection with SARS-CoV-2 can lead to significant multi-organ dysfunction, including cardiovascular, pulmonary, neurologic, and musculoskeletal systems [2]. This can result in physical disability, particularly among those who required hospitalization.

There is growing evidence that COVID-19 leads to long-term deficits in physical function. For example, a recent analysis of over 135,000 COVID-19 patients from the US Department of Veterans Affairs found a high burden of disability and health loss up to 3 years after hospitalization [2]. Similarly, a systematic review conducted in 2024 of 106 studies examining physical function recovery following acute COVID-19 illness found persistent impairments in physical function over 11 months of follow-up [3]. Despite these findings, significant gaps remain. Only two studies assessed physical function up to 24 months after hospitalization, and neither captured pre-illness data from participants [4,5]. This omission limits the ability to determine whether observed impairments are new or pre-existing. Furthermore, few studies comprehensively examine a broad range of recovery domains—such as mobility, cognition, mental health, and quality of life—within the same cohort, particularly over extended follow-up periods.

In a multi-center prospective cohort study, we recently reported that 55−60% of patients previously hospitalized for COVID-19 had clinically important deficits in mobility and physical performance compared to their pre-illness status that persisted up to 12 months after hospital discharge [6]. In this paper, we report the results of a follow-up study examining functional recovery 24-months after hospitalization for COVID-19. Specifically, we aimed to examine 1) the 24-month effects of hospitalization for COVID-19 on mobility, cognition, symptoms, mental health and health-related quality of life and 2) predictors of 24-month functional recovery in this population. We hypothesized that mobility disability would persist at 24-month follow-up when compared to pre-morbid status.

## Materials and methods

### Study design

The COREG-FR was a prospective cohort study (NCT04602260), which enrolled patients hospitalized for COVID-19 across five Ontario hospitals, Canada

between August 21, 2020 and December 21, 2021 [6,7]. This period encompassed the first four waves of the pandemic in Canada, which included multiple variants of concern (Alpha, Beta, Delta, Gamma and Omicron) [8]. Methods of the COREG-FR study and results over 12-months of follow-up were reported elsewhere [6,7]; key methods for the 24-month extension study are described below. This study was approved by the Hamilton Integrated Research Ethics Board (HIREB #11049) and the Waterloo-Wellington Research Ethics Board (WWREB 2020-0710).

## Study participants

We enrolled adult patients (≥18 years) who were either currently or recently hospitalized for a COVID-19 infection, as defined by the International Severe Acute Respiratory and Emerging Infection Consortium. Using daily Infection Prevention and Control data, site leads identified potential patients admitted to a medical unit, emergency department or intensive care unit based on either a confirmed positive nasopharyngeal swab or a documented COVID-19 diagnosis. Patients were excluded if they previously resided in an institution (e.g., long term care facility), had severe premorbid physical functional limitations (i.e., unable to stand independently), or had cognitive impairment which would limit their ability to complete follow-up assessments. Verbal informed consent was obtained from all participants.

## Study procedures

A research assistant contacted all patients who completed the COREG-FR 12-month follow-up assessment and invited them to participate in a 24-month follow-up study. Those who consented completed a telephone interview with research staff lasting between 30 and 45 minutes 24-months after their hospital discharge.

## Outcome measures

Participant demographics (e.g., age, sex), health information (e.g., comorbidities) and information about hospitalization (e.g., intensive care unit admission, length of stay) were collected using the COREG registry [7]. The primary outcome was the Activity Measure for Post-Acute Care (AM-PAC) Basic Mobility Domain. As a secondary outcome, we used the AM-PAC Cognition Domain to assess applied cognition [9]. The AM-PAC is a patient- or proxy-reported or clinician-administered outcome measure that has been validated for post-acute care settings [9,10]. The AM-PAC has been shown to be more responsive to change than the Functional Independent Measure [10,11]. Multiple short-forms of the AM-PAC exist (e.g., AM-PAC Inpatient 6 Clicks & AM-PAC Outpatient) depending on the setting [11–13]. Short-form scores from each domain are converted to a standardised score which allow comparison across the different AM-PAC versions. Items are scored from 1 (unable to perform) to 4 (none or no difficulty) with higher overall scores indicating better function. The AM-PAC Outpatient Short-Form Basic Mobility and Cognition Domains were assessed close to the time of hospital admission (asking about premorbid status) and every three months after hospital discharge until the 12-month follow-up, with an additional assessment at 24 months. The persistence of mobility deficits at 24 months was determined using the minimal clinically important difference (MCID) in mobility (MCID ≥ 3.3) [14] and applied cognition (MCID ≥ 5.5) [15] in reference to premorbid levels. To further characterize recuperation, at each follow-up timepoint, we asked participants how much their COVID-19 recovery continued to affect their normal daily activities within the preceding week; response options ranged from 'not at all' to 'all the time'.

Secondary outcomes included symptoms of COVID-19 assessed using the Fatigue Visual Analogue Scale (VAS) [16] and the Medical Research Council (MRC) dyspnea scale [17]. Fatigue VAS scores range from 0 to 10 with lower scores indicating worse global fatigue [16]. MRC dyspnea scale includes five statements which describe the extent of breathlessness from breathless with strenuous exercise to being too breathless to leave the house [17]. Scores range from 1 to 5 with higher scores indicating more severe breathlessness [17].

Mental health outcomes were also assessed using the Impact of Event Scale-Revised (IES) [18] and the Hospital Anxiety and Depression Scale (HADS) [19]. The IES is a self-report measure which assesses subjective distress due to traumatic events [18]. It is comprised of 22 items with each item scored on a 5-point scale from 0 ("not at all") to 4 ("extremely") [18]. Overall scores range from 0 to 88 with higher scores indicating worse distress [18]. Subscales of the IES are Intrusion, Avoidance and Hyperarousal [18]. The measurement properties of the IES demonstrate strong validity for assessing trauma-related distress [20,21].

The HADS is a self-report measure which contains 14-items, with seven relating to the anxiety domain and seven pertaining to the depression domain [19]. Each item is scored from 0 to 3 with overall domain sores ranging from 0 to 21 [19]. Higher scores on the anxiety domain indicate more severe anxiety [19] while a cut-off of ≥8 on the depression domain indicates depression among a general population [19].

Health-related quality of life was assessed using the EuroQoL-5D-5L (EQ-5D-5L) and its VAS [22]. The EQ-5D-5L assesses five domains (mobility problems, self-care problems, usual activities problems, pain/discomfort, and anxiety/depression) with 5 response options ranging from "no problems" to "extreme problems or unable to" [23]. The VAS characterizes overall health with scores ranging from 0 (worst imaginable health state) to 100 (best imaginable health state) [22]. The EQ-5D-5L has excellent measurement properties across a broad range of ill populations [24].

## Statistical methods

Nominal and ordinal variables are reported as counts and percentages. Continuous variables are reported as mean (standard deviation [SD]) if normally distributed or median (1st - 3rd quartiles) if non-normally distributed.

The proportion of patients experiencing clinically important deficits in mobility and cognition at 24 months compared to their premorbid status was evaluated using the AM-PAC mobility and cognition MCID values [14,15]. One-way repeated measures analysis of variance (ANOVA) was conducted to determine whether there were significant differences in mobility and applied cognition scores between premorbid, 12 months, and 24 months. In the case of a significant ANOVA result, Tukey's 95% family-wise confidence intervals for multiple comparisons were used to identify the locus of the differences.

We assessed the predictors of 24-month functional recovery (AM-PAC Mobility score) using multiple linear regression. Predictors included: hospital length of stay, sex, age, number of comorbidities, general and mental health, percent predicted forced expiratory volume in 1 second (FEV1) at 3 months, household income, education level, and pre-morbid mobility [25,26]. In the first model, we assessed the association of all predictors (except pre-morbid mobility) with 24-month functional recovery. The second model assessed the association of pre-morbid mobility with 24-month functional recovery. Last, we assessed the association of all predictors (including pre-morbid mobility) with 24-month functional recovery. All variables were included in the final model. A p-value less than 0.05 was considered to be statistically significant. After confirming FEV1 scores improved modeling via complete case analysis, we used multiple imputation by predictive mean matching to manage missing FEV1 scores at 3 months. Regression analyses were conducted using R (version 4.4.2) with all other statistical analyses conducted using Stata (16.1, StataCorp LLC, College Station, TX) with an alpha level of 0.05.

## Results

Of the 215 participants who completed the previously published 12-month follow-up [6], 170 (79%) consented to the 24-month follow-up interview (Fig 1). Twelve-month follow-up participants with at least post-secondary education were more likely to consent to the 24-month follow-up; no consent differences were observed by sex, age, household income, hospital length of stay, baseline AM-PAC mobility or cognitive scores. The mean age at 24-month follow-up was 61.2 (12.7) years, with 91 (53.5%) being male (Table 1). At the 24-month follow-up, 16% and 18% of patients experienced symptoms of anxiety and depression, respectively (Table 2). Additionally, 38% of patients reported mobility problems, such as walking difficulties, and 51% continued to experience pain or discomfort.

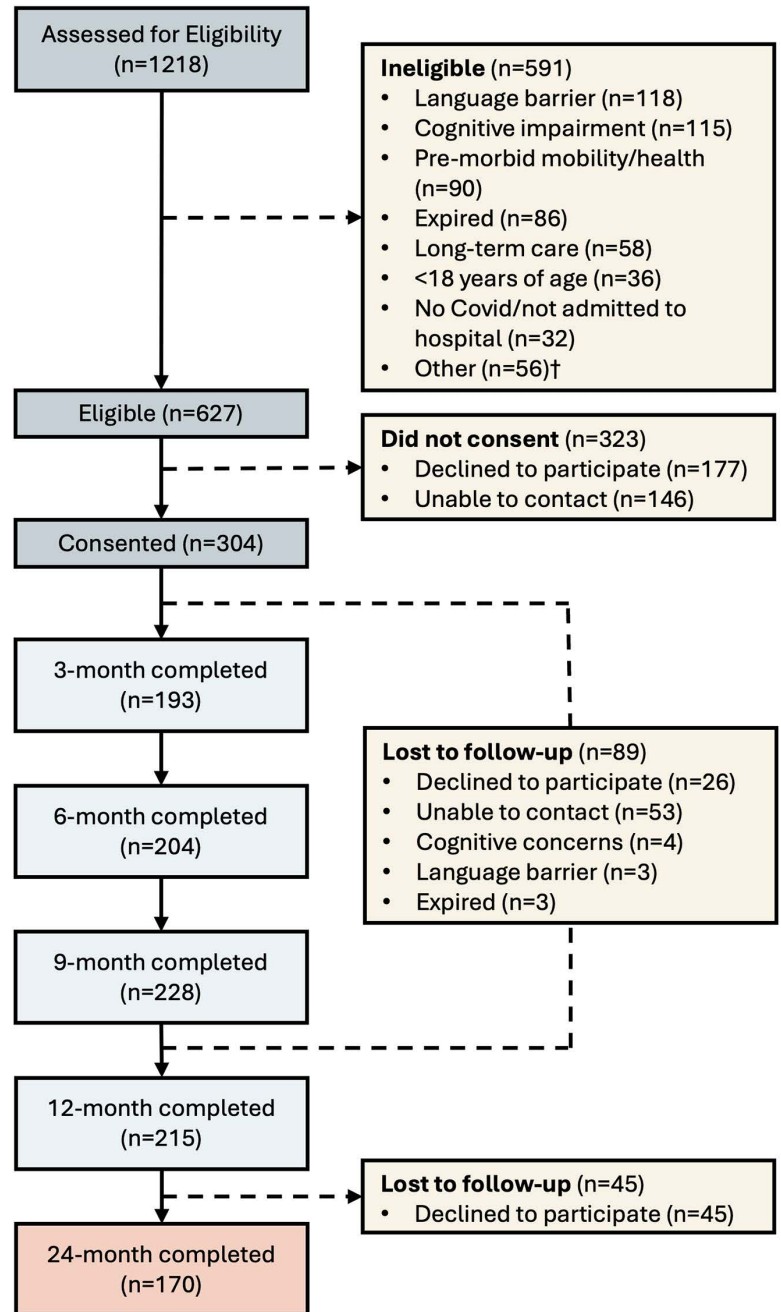

**Fig 1. Participant flowchart.** † Other reasons for ineligibility included admission to ICU only, hearing impairment, substance abuse homeless, prisoner, enrolled in another study.

Fig 2 shows the proportion of patients with clinically important deficits in AM-PAC mobility (57%) and cognition (41%) scores, and who reported that their COVID-19 continues to impact their daily activities (59%) at the 24-month follow-up. Compared to premorbid levels, AM-PAC mobility scores remained significantly lower at both 12 months (p < 0.001) and 24 months (p < 0.001), with no significant difference between the 12-month and 24-month scores (p = 0.181) (Fig 3). Similarly,

**Table 1. Demographics of patients at 24-months follow-up.**

|  | N = 170 |
|---|---|
| **Patient characteristics** |  |
| Age in years, mean (SD) | 61.2 (12.7) |
| Range years | 32–92 |
| Sex |  |
| Males (%) | 91 (54) |
| Females (%) | 79 (46) |
| Comorbidities, N (%) |  |
| <3 | 101 (59) |
| ≥3 | 69 (41) |
| High blood pressure | 72 (42) |
| Diabetes | 51 (30) |
| Cancer | 25 (15) |
| COPD | 21 (12) |
| Hospital length of stay days, mean (SD) | 13.9 (19.6) |
| Median (1st - 3rd quartiles) | 8 [4–15] |
|  | n = 142 |
| Admission to ICU, N (%) | 49 (35) |
| Length of stay days in ICU, mean (SD) | 14 (16) |
| Median [IQR] | 8 [4–18] |
| **COVID-19 Vaccines, N (%)** |  |
| Have you received at least one dose of a COVID-19 vaccine? |  |
| Yes | 151 (89) |
| No | 19 (11) |
| How many doses of the COVID-19 vaccine have you received so far? |  |
| 1 dose | 4 (3) |
| 2 doses | 30 (20) |
| 3 or more | 117 (77) |
| **In the past week, how much has COVID-19 and your recovery from this illness affected your normal daily activities? N (%)** |  |
| Not at all | 69 (40) |
| A little bit | 60 (35) |
| Somewhat | 17 (10) |
| Quite a bit | 16 (9) |
| All the time | 8 (5) |
| **Health, N (%)** |  |
| General Health |  |
| Excellent or very good | 40 (24) |
| Good | 71 (42) |
| Fair or Poor | 59 (35) |
| Mental Health |  |
| Excellent or very good | 37 (22) |
| Good | 83 (49) |
| Fair or Poor | 49 (29) |

**Table 2. Results at 3-, 6-, 9-, 12- and 24-months follow-up after hospital discharge for physical and mental health measures for COVID-19 survivors.**

| Symptom severity | 3 months | 6 months | 9 months | 12 months | 24 months |
|---|---|---|---|---|---|
| **Fatigue** | | | | | |
| VAFS<br>Median [1st - 3rd quartiles] | N=124<br>70 [50–87.5] | N=144<br>75 [50–90] | N=70<br>70 [50–90] | N=161<br>70.0 [60–90] | N=170<br>80 [60–100] |
| **Dyspnea** | | | | | |
| mMRC scale scores for dyspnea, N (%) | N=128 | N=147 | N=165 | N=163 | N=170 |
| 0 | 40 (31) | 61 (42) | 65 (39) | 69 (42) | 79 (46) |
| ≥1 | 88 (69) | 86 (59) | 100 (61) | 94 (58) | 91 (54) |
| **Psychological Assessment** | | | | | |
| HADS Anxiety<br>score ≥ 8, N (%) | N=124<br>33 (27) | N=143<br>40 (28) | N=162<br>45 (28) | N=161<br>35 (22) | N=168<br>27 (16) |
| HADS Depression<br>score ≥ 8, N (%) | N=122<br>33 (27) | N=142<br>38 (27) | N=162<br>42 (26) | N=160<br>37 (23) | N=170<br>30 (18) |
| **Health Status** | | | | | |
| EQ-5D-5L (score ≥2)<br>Mobility problems walking around, N (%) | N=124<br>61 (49) | N=144<br>69 (48) | N=162<br>80 (49) | N=160<br>75 (47) | N=170<br>64 (38) |
| Self-care problems with washing or dishing, N (%) | 25 (20) | 22 (15) | 23 (14) | 28 (18) | 23 (14) |
| Usual Activities problems, N (%) | 64 (52) | 55 (38) | 80 (49) | 79 (49) | 69 (41) |
| Pain/Discomfort, N (%) | 81 (65) | 89 (62) | 93 (57) | 90 (56) | 87 (51) |
| Anxiety/Depression, N (%) | 60 (48) | 51 (35) | 78 (48) | 75 (47) | 67 (39) |
| Quality of life<br>median [1st - 3rd quartiles] | 75 [60–85] † | 75 [60–85] ‡ | 72 [60–84] § | 75 [60–85] | 75 [65–85] |

*Legend:* VAFS: Visual Analogue Fatigue Scale from 0 (worst) to 100 (normal); mMRC: Modified Medical Research Council scores from 0 (no breathlessness except on strenuous exercise) to 1–4 (shortness of breath when walking); HADS Hospital and Anxiety Depression Scale scores ≥ 8 borderline/pathological anxiety and depression; EQ-5D-5L: 5-level EuroQol score ≥ 2 any problem; Quality of Life assessed using the EQ-5D-5L visual analogue scale from 0 (worst imaginable health state) to 100 (best imaginable health state); † N = 123; ‡ N = 143; § N = 161.

AM-PAC applied cognition scores were significantly below premorbid scores, with no difference between the 12-month and 24-month scores (p = 0.534) (Fig 3).

Multiple linear regression demonstrated that having better mobility recovery at 24-months was associated with being younger and male, having fewer comorbidities, better premorbid general health, a higher FEV1% predicted 3-months post-illness, and a household income of at least $50,000 (Table 3). In the hierarchical entry model, better premorbid mobility was significantly associated with higher recovery at 24-months (beta coefficient and 95% confidence interval = 0.49 [0.37, 0.62]) along with having fewer comorbidities (−1.12 [−1.58, −0.66]), a household income of at least $50,000 (2.37 [0.48, 4.26]) and a higher FEV1% predicted at 3-months (0.09 [0.01, 0.18]) (Table 3).

## Discussion

We identified that 57% and 41% of patients continue to have clinically important deficient in mobility and cognition, respectively, at 24-month follow-up. These findings highlight the ongoing challenges faced by patients recovering from COVID-19, even two years post-hospitalization. The persistence of clinically important deficits in mobility and cognition coupled with high levels of anxiety or depression and persistent physical discomfort, reflects the profound and enduring effect of COVID-19 on functional health for a substantial proportion of patients.

Our results demonstrate stability in the proportion of COVID-19 survivors with mobility and cognitive deficits between 12- and 24-months after hospitalization. This recovery plateau indicates that usual care may be insufficient to support

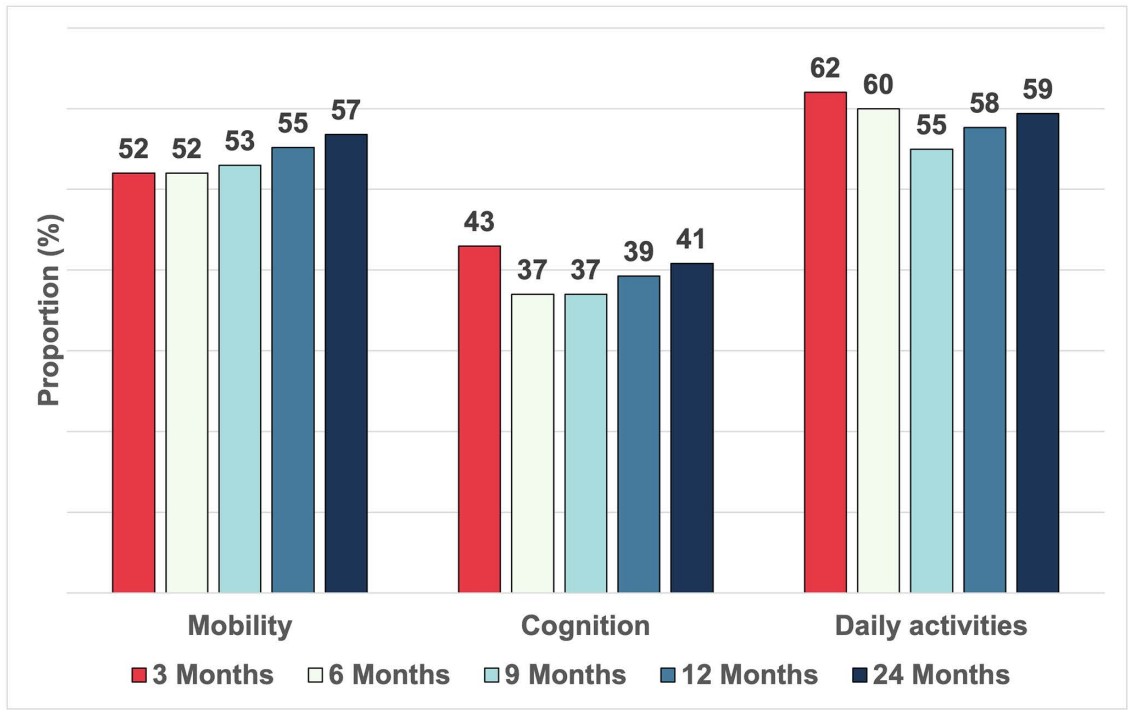

**Fig 2. The proportion of patients with clinically important mobility deficits (3 Months N = 128; 6 months N = 147; 9 Months N = 165; 12 months N = 163; 24 months N = 169), clinically important cognition deficits (3 Months N = 128; 6 months N = 147; 9 Months N = 165; 12 months N = 163; 24 months N = 169), and COVID-19 recovery impacting their daily activities by follow-up timepoints (3 Months N = 125; 6 months N = 144; 9 Months N = 165; 12 months N = 163; 24 months N = 170) at 12 and 24 months.**

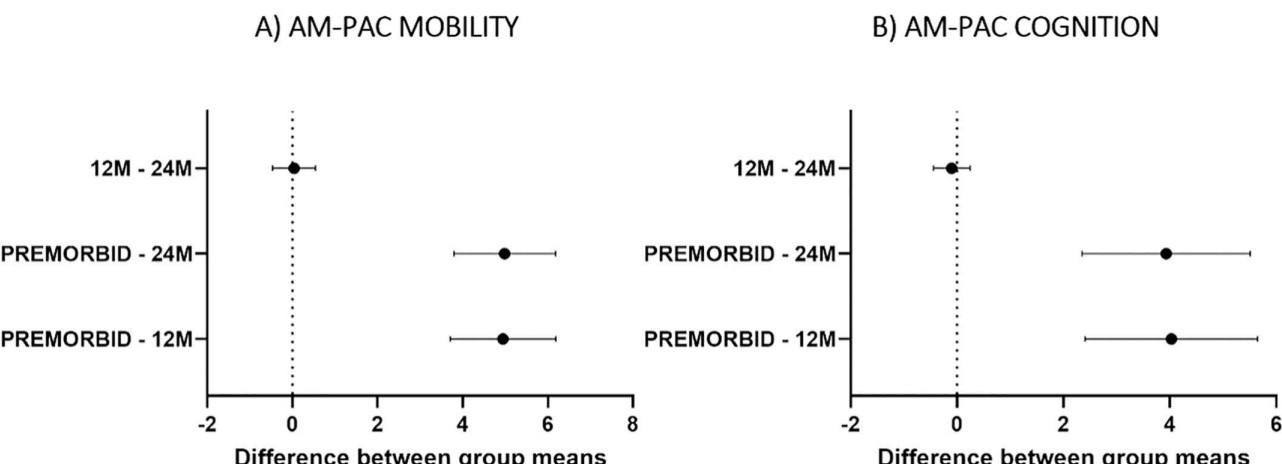

**Fig 3. Simultaneous 95% confidence interval of the differences in the mean t score of the AM-PAC basic mobility (A) and applied cognition (B) across assessment timepoints (premorbid, 12-month follow-up and 24-month follow-up).** If an interval does not contain zero, the corresponding pair of groups' means (vertical axis) are significantly different. Positive confidence intervals (horizontal axis) indicate worse mobility or cognition at the later timepoint (N = 169).

**Table 3. Summary of regression analyses for the predictors of mobility at 24-month follow-up.**

| AM-PAC Mobility Score at 24-Month Follow-Up | β | SE β | 95% CI for β | | P | R | R² | Δ R² |
|---|---|---|---|---|---|---|---|---|
| | | | LLsec | UL | | | | |
| **Model 1** | | | | | | 0.76 | 0.58 | 0.56 |
| Intercept | 64.23 | 4.48 | 55.183 | 73.29 | **<0.001** | | | |
| Hospital LOS (days) | 0.00 | 0.01 | −0.01 | 0.02 | 0.612 | | | |
| Sex (female) | −3.82 | 1.07 | −5.93 | −1.70 | **0.001** | | | |
| Age (years) | −0.14 | 0.05 | −0.23 | −0.05 | **0.002** | | | |
| Number of comorbidities | −1.40 | 0.27 | −1.93 | −0.88 | **<0.001** | | | |
| General Health (very good, excellent) | 3.52 | 1.32 | 0.93 | 6.10 | **0.008** | | | |
| FEV1 (% Predicted) at 3 months | 0.11 | 0.04 | 0.02 | 0.20 | **0.018** | | | |
| Household income (≥ $50,000) | 3.90 | 1.10 | 1.74 | 6.06 | **0.004** | | | |
| Education (≥ some post-secondary) | 1.69 | 1.10 | −0.47 | 3.85 | 0.126 | | | |
| Mental Health (very good, excellent) | 0.29 | 1.40 | −2.46 | 3.05 | 0.835 | | | |
| **Model 2** | | | | | | 0.75 | 0.56 | 0.55 |
| Intercept | 9.79 | 3.80 | 2.35 | 17.23 | **0.001** | | | |
| Premorbid mobility | 0.79 | 0.05 | 0.68 | 0.89 | **<0.001** | | | |
| **Model 3** | | | | | | 0.83 | 0.69 | 0.67 |
| Intercept | 27.70 | 6.14 | 15.59 | 39.82 | **<0.001** | | | |
| Premorbid mobility | 0.49 | 0.06 | 0.37 | 0.62 | **<0.001** | | | |
| Hospital LOS (days) | 0.00 | 0.01 | −0.01 | 0.01 | 0.857 | | | |
| Sex (female) | −1.72 | 0.96 | −3.61 | 0.17 | 0.074 | | | |
| Age (years) | −0.05 | 0.04 | −0.13 | 0.03 | 0.200 | | | |
| Number of comorbidities | −1.12 | 0.23 | −1.58 | −0.66 | **<0.001** | | | |
| General Health (very good, excellent) | 1.64 | 1.17 | −0.66 | 3.94 | 0.162 | | | |
| Household income (≥ $50,000) | 2.37 | 0.96 | 0.48 | 4.26 | **0.014** | | | |
| Education (≥ some post-secondary) | 0.49 | 0.950 | −1.38 | 2.36 | 0.605 | | | |
| FEV1 (% Predicted) at 3 months | 0.09 | 0.04 | 0.01 | 0.18 | **0.031** | | | |
| Mental Health (very good, excellent) | 1.06 | 1.21 | −1.31 | 3.43 | 0.380 | | | |

*Legend:* Significant *p* values identified in bold font. β = beta coefficient; SE β = standard error of beta coefficient; CI = confidence interval; R = correlation coefficient; R² = R-square or the coefficient of determination; Δ R² = adjusted R²; LL = lower limit; UL = upper limit.

patients to return to their pre-illness levels of functioning after a COVID-19 infection and highlights the need for ongoing monitoring and rehabilitative therapies tailored to patients' persistent issues. Similar to our 12-month report, better premorbid mobility, fewer comorbidities, higher FEV1 scores at 3-months and a household income of at least $50,000 predicted better recovery at 24-months.

A previous longitudinal cohort study of 1,192 survivors of hospitalization for COVID-19 found that at 24-months, mean 6-minute walk test (6MWT) distance was 94% (84.7 to 104.1) of predicted, with only 8% of participants falling below the lower limit of normal [4]. This contrasts with our results which found that deficits in self-reported physical function persisted 24-months after hospitalization. The differing findings are likely related to differences in sample characteristics (e.g., median age of 57 years), outcome measures, and importantly, the lack of pre-illness walk score in the previous study which makes comparisons difficult. Notably, the 6MWT score is a performance-based measure in which participants walk as far as possible for 6 minutes (i.e., a measure of capacity). Conversely, the AM-PAC is a patient-reported measure [9] that asks participants to rate their physical function in daily tasks, beyond walking.

A recent systematic review examined the persistence of post-COVID-19 symptoms among studies which enrolled both previously hospitalized and non-hospitalized COVD-19 survivors [27]. Within their meta-analysis of 2,837 participants,

they found that 28% (1st-3rd quartiles = 13–46%) of survivors continued to have cognitive deficits 2 years after infection. This is a somewhat lower prevalence than our study in which nearly 40% experienced cognitive deficits at 2 years, however the systematic review reported on memory and concentration constructs, while the AM-PAC applied cognitive domain includes assessments of communication, print information, new learning and social cognition. Nonetheless, these data suggest the importance of long-term cognitive deficits in this population.

This study has limitations. The reliance on self-reported data for both premorbid and follow-up assessments may introduce bias, potentially affecting the perceived severity of deficits. However, we used standardized and well-validated outcome measures for all primary and secondary outcomes. Additionally, there is a potential for selection bias, as individuals experiencing persistent symptoms may have been more likely to participate in the 24-month follow-up; however, our comparison of patients who consented versus those who declined did not reveal any substantive differences. While the sample size was limited to 170 participants, this study, to our knowledge, is the first to report repeated measures of physical function and cognition from premorbid status through 2 years post-hospitalization in survivors of COVID-19.

## Conclusions and implications

Functional and cognitive deficits after moderate to severe COVID-19 do not meaningfully resolve over 2 years post-infection for over 40% of patients. There is a need for early interventions and rehabilitation to prevent long-term functional challenges in hospitalized COVID-19 patients. A strong emphasis should be placed on developing comprehensive rehabilitation programs tailored specifically for recovering COVID-19 patients that target mobility, cognition, mental health and symptom management.

## Author contributions

**Conceptualization:** Marla Beauchamp.

**Formal analysis:** Marla Beauchamp, Christopher Farley, Renata Kirkwood, Aaron Jones, Parminder Raina, Jinhui Ma.

**Funding acquisition:** Marla Beauchamp.

**Investigation:** Marla Beauchamp, Renata Kirkwood.

**Methodology:** Marla Beauchamp, Christopher Farley, Renata Kirkwood, Terence Ho, MyLinh Duong, Jinhui Ma.

**Project administration:** Marla Beauchamp.

**Resources:** Marla Beauchamp.

**Supervision:** Marla Beauchamp, Jinhui Ma.

**Validation:** Marla Beauchamp, Renata Kirkwood.

**Visualization:** Christopher Farley, Renata Kirkwood.

**Writing – original draft:** Marla Beauchamp, Christopher Farley, Renata Kirkwood, Jinhui Ma.

**Writing – review & editing:** Marla Beauchamp, Christopher Farley, Renata Kirkwood, Aaron Jones, Terence Ho, MyLinh Duong, Parminder Raina, Jinhui Ma.

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
