## [Decision Letter · Decision Letter 0]

14 Jul 2025

Dear Dr. Beauchamp,

Thank you for submitting your manuscript to PLOS ONE. After careful consideration, we feel that it has merit but does not fully meet PLOS ONE’s publication criteria as it currently stands. Therefore, we invite you to submit a revised version of the manuscript that addresses the points raised during the review process.

We look forward to receiving your revised manuscript.

Kind regards,

Armaan Jamal

Guest Editor

PLOS ONE

Journal Requirements:

The COREG Registry and COREG-FR are funded by a Canadian Institutes of Health Research grant (172754). The COREG Registry is also funded by a Hamilton Academic Health Sciences Organisation grant (HAH-21−04). The funders had no role in study design, data collection and analysis, decision to publish, or preparation of this manuscript.

The COREG Registry and COREG-FR are funded by a Canadian Institutes of Health Research grant (172754). The COREG Registry is also funded by a Hamilton Academic Health Sciences Organisation grant (HAH-21−04). The funders had no role in study design, data collection and analysis, decision to publish, or preparation of this manuscript.

3. In the online submission form, you indicated that data are available upon request to Dr. Marla Beauchamp (beaucm1@mcmaster.ca ).

Reviewers' comments:

Reviewer's Responses to Questions

**Comments to the Author**

1. Is the manuscript technically sound, and do the data support the conclusions?

Reviewer #1: Yes

Reviewer #2: Partly

Reviewer #3: Yes

Reviewer #4: Yes

Reviewer #5: Partly

2. Has the statistical analysis been performed appropriately and rigorously?

Reviewer #1: Yes

Reviewer #2: Yes

Reviewer #3: Yes

Reviewer #4: Yes

Reviewer #5: Yes

3. Have the authors made all data underlying the findings in their manuscript fully available?

Reviewer #1: Yes

Reviewer #2: Yes

Reviewer #3: Yes

Reviewer #4: No

Reviewer #5: No

4. Is the manuscript presented in an intelligible fashion and written in standard English?

Reviewer #1: Yes

Reviewer #2: Yes

Reviewer #3: Yes

Reviewer #4: Yes

Reviewer #5: Yes

Reviewer #1: The presented study is a 24-month cohort and shows the evolution of physical and cognitive function in patients who had moderate to severe COVID-19.

It is written clearly and is easy to understand, with a well-contextualized introduction focused on the study's objective.

The assessment tools are described in the methods section and are easily replicable, which positively contributes to the publication of an article.

Regarding the results, they are well demonstrated in the tables and graphs and appropriately show the evolution of the patients' functional and cognitive status.

Figure 3 needs to be better explained regarding the 12-month and 24-month premorbidity.

In the results, it is clear what happens when the difference is zero, but it is not as clear concerning the other data in the graph.

The discussion is methodologically appropriate and makes comparisons with similar studies.

Reviewer #2: This manuscript requires a significant amount of improvement in:

Explain the novelty of the study

a. What is known about the topic? (Background)

b. What is not known? (The research problem)

c. Why the study was done? (Justification)

Introduction

Background/rationale (Explain the scientific background and rationale for the investigation being reported)

Objectives (State-specific objectives, including any prespecified hypotheses)

Methods

Data sources/ measurement

For each variable of interest, give sources of data and details of methods of assessment (measurement). Describe comparability of assessment methods

Variables

Clearly define all outcomes, exposures, predictors, potential confounders, and effect modifiers. Give diagnostic criteria

Bias

Describe any efforts to address potential sources of bias

Study size

Explain how the study size was arrived at

Discussion

Summarise key results with reference to study objectives

Discuss limitations of the study, taking into account sources of potential bias or imprecision. Discuss both direction and magnitude of any potential bias

Reviewer #3: Thank you for the invitation to review this article. It is an exciting article to publish. The title is Functional recovery 2-years after hospitalization for COVID-19: Insights from the COREG-FR extension study, and the ID number is (PONE-D-25-16167).

Reviewer #4: This manuscript is clear, concise, and seems well-analyzed. The limitations of the study are clearly stated. It's always good to see the protocol was registered ahead of time! As far as I can tell, all the statistics were done rigorously. However, I am concerned with the data availability statement (or lack thereof). The authors have not explained what exceptional circumstances are present to avoid data availability. There should also be some sort of institutional contact for data availability according to PLOS's data guidelines, rather than just an email address for an author. I understand that clinical data probably can't be released in full, but the factors that go into this decision need to be more fully explained.

As a small note, I found the first data point in Table 1 was confusing to the reader. Perhaps "Age in years, mean (SD)" would make it more clear what value was included in the parentheses inside the table.

Reviewer #5: I appreciate the opportunity to review this paper. The paper is about the functional recovery following COVID-19 infection, focusing on the 2-year period. Although the topic is interesting, the paper does not seem to add to the current literature. Main findings of the study, such as the positive association between higher income and lack of comorbidity with better outcomes, are already well established. The data is old, and the study population is very limited. Following, you can find some suggestions that may help to improve the paper:

- The abstract requires major revision. Please add some background to the introduction section of the abstract. The method can be much shorter. Please remove unnecessary components and paraphrase the sentences to shorten them. Mention the time period for data collection in the method in the abstract.

- You should add the limited number of study population to the limitations of the study.

**Do you want your identity to be public for this peer review?** For information about this choice, including consent withdrawal, please see our Privacy Policy

Reviewer #1: No

Reviewer #2: No

Reviewer #3: No

Reviewer #4: No

Reviewer #5: No

---

## [Author Response · Author response to Decision Letter 1]

10 Sep 2025

Dear Dr. Armaan Jamal,

Thank you for reviewing our manuscript entitled: “Functional recovery 2-years after hospitalization for COVID-19: Insights from the COREG-FR extension study”. We are grateful for the external expert review and the opportunity to improve our work.

Below, please find a point-by-point response to each of the comments.

Editor’s Comments to Author:

Editor: When submitting your revision, we need you to address these additional requirements.

Response: Thank you for highlighting the formatting requirements. We have ensured our submission is consistent with the PLOS ONE style requirements.

Editor: 2. Thank you for stating the following in the Acknowledgments Section of your manuscript:

The COREG Registry and COREG-FR are funded by a Canadian Institutes of Health Research grant (172754). The COREG Registry is also funded by a Hamilton Academic Health Sciences Organisation grant (HAH-21−04). The funders had no role in study design, data collection and analysis, decision to publish, or preparation of this manuscript.

The COREG Registry and COREG-FR are funded by a Canadian Institutes of Health Research grant (172754). The COREG Registry is also funded by a Hamilton Academic Health Sciences Organisation grant (HAH-21−04). The funders had no role in study design, data collection and analysis, decision to publish, or preparation of this manuscript.

Response: Thank you for bringing this oversight to our attention. We have removed the funding statement from our acknowledgement section. The current Funding Statement on record is correct and requires not further modifications.

Editor: 3. In the online submission form, you indicated that data are available upon request to Dr. Marla Beauchamp (beaucm1@mcmaster.ca).

Response: We have now included the following statement. “Data cannot be shared publicly because of privacy concerns (i.e., sensitive patient information). While our research ethics board approval precludes making the dataset publicly available, summary data can be made available upon request (email: beaucm1@mcmaster.ca).”

Editor: 4. If the reviewer comments include a recommendation to cite specific previously published works, please review and evaluate these publications to determine whether they are relevant and should be cited. There is no requirement to cite these works unless the editor has indicated otherwise.

Response: Thank you for this comment. The reviewers have not suggested citing any further publications.

Editor: 5. Please review your reference list to ensure that it is complete and correct. If you have cited papers that have been retracted, please include the rationale for doing so in the manuscript text, or remove these references and replace them with relevant current references. Any changes to the reference list should be mentioned in the rebuttal letter that accompanies your revised manuscript. If you need to cite a retracted article, indicate the article’s retracted status in the References list and also include a citation and full reference for the retraction notice.

Response: Thank you for highlighting our reference list. We have revised our current reference list and believe it is in line with PLOS ONE guidelines; additionally, we have not cited any retracted publications.

Reviewers' comments to author:

REVIEWER 1:

R1: The presented study is a 24-month cohort and shows the evolution of physical and cognitive function in patients who had moderate to severe COVID-19.

It is written clearly and is easy to understand, with a well-contextualized introduction focused on the study's objective.

The assessment tools are described in the methods section and are easily replicable, which positively contributes to the publication of an article.’

Regarding the results, they are well demonstrated in the tables and graphs and appropriately show the evolution of the patients' functional and cognitive status.

Response: Thank you for this positive feedback regarding our manuscript.

R1: Figure 3 needs to be better explained regarding the 12-month and 24-month premorbidity.

In the results, it is clear what happens when the difference is zero, but it is not as clear concerning the other data in the graph.

Response: Thank you for highlighting this issue. To aid interpretation, we have revised the legend of Figure 3 as follows starting on line 262:

“Simultaneous 95% confidence interval of the differences in the mean t score of the AM-PAC basic mobility (A) and applied cognition (B) across assessment timepoints (premorbid, 12-month follow-up and 24-month follow-up). If an interval does not contain zero, the corresponding pair of groups’ means (vertical axis) are significantly different. Positive confidence intervals (horizontal axis) indicate worse mobility or cognition at the later timepoint (N=169).”

R1: The discussion is methodologically appropriate and makes comparisons with similar studies.

Response Thank you for your constructive feedback to improve the clarity of our manuscript.

REVIEWER 2:

R2: This manuscript requires a significant amount of improvement in:

Explain the novelty of the study

a. What is known about the topic? (Background)

b. What is not known? (The research problem)

c. Why the study was done? (Justification)

Response: Thank you for reviewing our manuscript. We have provided additional detail below to clarify how our study builds on and extends the existing literature.

a. What is known about the topic (Background):

There is growing evidence that COVID-19 leads to long-term deficits in physical function. For example, a recent analysis of over 135,000 COVID-19 patients from the US Department of Veterans Affairs reported a substantial burden of disability and health loss lasting up to 3 years post-hospitalization. Similarly, a 2024 systematic review of 106 studies examining physical function recovery following acute COVID-19 illness found persistent impairments in physical function for up to 11 months after discharge. This body of evidence confirms the long-term impact of COVID-19 on functional outcomes.

b. What is not known (The research problem):

Despite these findings, significant gaps remain. Only two studies to date have assessed physical function outcomes at 24 months post-hospitalization, and neither included pre-illness functional data. This omission limits the ability to determine whether observed impairments are new or pre-existing. Furthermore, few studies comprehensively examine a broad range of recovery domains—such as mobility, cognition, mental health, and quality of life—within the same cohort, particularly over extended follow-up periods.

c. Why the study was done (Justification):

Our study was designed to address these critical gaps in the literature. Using data from a multi-center prospective cohort, we tracked patients previously hospitalized for COVID-19 and assessed their functional status at 24 months post-discharge, comparing it to their pre-illness baseline. By incorporating premorbid data and evaluating multiple domains of recovery—including mobility, cognition, symptoms, mental health, and quality of life—we provide a more comprehensive and accurate assessment of long-term outcomes. This approach allows us to identify not only the persistence of disability but also the factors that may predict or hinder recovery. To our knowledge, this is the first study with 24-month follow-up after COVID-19 hospitalization that includes baseline (pre-illness) function, offering unique insight into true recovery trajectories.

We have clarified and expanded these points in the revised manuscript (see lines 65–76).

R2: Introduction

Background/rationale (Explain the scientific background and rationale for the investigation being reported)

Response: We have identified background knowledge from lines 65 to 76. This includes synthesis from a systematic review which identified physical disability at 2 years after hospitalization. However, as we identify on line 66, no study with 2-year follow-up includes premorbid data, which is a significant limitation to the current state of knowledge that our paper addresses. For clarity, we have expanded on this point in our response above.

R2: Objectives (State-specific objectives, including any prespecified hypotheses)

Response: Thank you for highlighting this oversight. We have revised our introduction to be more explicit about our objectives (starting on line 81) and added our hypotheses starting on line 91 as follows:

“We hypothesized that mobility disability would persist at 24-month follow-up when compared to pre-morbid status.”

R2: Methods

Data sources/ measurement

For each variable of interest, give sources of data and details of methods of assessment (measurement). Describe comparability of assessment methods

Response: Thank you for identifying this oversight. We have provided extensive revisions to our methods section to more comprehensively reflect our approach starting on line 126:

“Participant demographics (e.g., age, sex), health information (e.g., comorbidities) and information about hospitalization (e.g., intensive care unit admission, length of stay) were collected using the COREG registry (7). The primary outcome was the Activity Measure for Post-Acute Care (AM-PAC) Basic Mobility Domain. As a secondary outcome, we used the AM-PAC Cognition Domain to assess applied cognition (9). The AM-PAC is a patient- or proxy-reported or clinician-administered outcome measure that has been validated for post-acute care settings (9, 10). The AM-PAC has been shown to be more responsive to change than the Functional Independent Measure (10, 11). Multiple short-forms of the AM-PAC exist (e.g. AM-PAC Inpatient 6 Clicks & AM-PAC Outpatient) depending on the setting (11-13). Short-form scores from each domain are converted to a standardised score which allow comparison across the different AM-PAC versions. Items are scored from 1 (unable to perform) to 4 (none or no difficulty) with higher overall scores indicating better function. The AM-PAC Outpatient Short-Form Basic Mobility and Cognition Domains were assessed close to the time of hospital admission (asking about premorbid status) and every three months after hospital discharge until the 12-month follow-up, with an additional assessment at 24 months. The persistence of mobility deficits at 24 months was determined using the minimal clinically important difference (MCID) in mobility (MCID ≥ 3.3) (14) and applied cognition (MCID ≥ 5.5) (15) in reference to premorbid levels. To further characterize recuperation, at each follow-up timepoint, we asked participants how much their COVID-19 recovery continued to affect their normal daily activities within the preceding week; response options ranged from ‘not at all’ to ‘all the time’.

Secondary outcomes included symptoms of COVID-19 assessed using the Fatigue Visual Analogue Scale (VAS) (16) and the Medical Research Council (MRC) dyspnea scale (17). Fatigue VAS scores range from 0 to 10 with lower scores indicating worse global fatigue (16). MRC dyspnea scale includes five statements which describe the extent of breathlessness from breathless with strenuous exercise to being too breathless to leave the house (17). Scores range from 1 to 5 with higher scores indicating more severe breathlessness (17).

Mental health outcomes were also assessed using the Impact of Event Scale-Revised (IES) (18) and the Hospital Anxiety and Depression Scale (HADS) (19). The IES is a self-report measure which assesses subjective distress due to traumatic events (18). It is comprised of 22 items with each item scored on a 5-point scale from 0 (“not at all”) to 4 (“extremely”) (18). Overall scores range from 0 to 88 with higher scores indicating worse distress (18). Subscales of the IES are Intrusion, Avoidance and Hyperarousal (18). The measurement properties of the IES demonstrate strong validity for assessing trauma-related distress (20, 21).

The HADS is a self-report measure which contains 14-items, with seven relating to the anxiety domain and seven pertaining to the depression domain (19). Each item is scored from 0 to 3 with overall domain sores ranging from 0 to 21 (19). Higher scores on the anxiety domain indicate more severe anxiety (19) while a cut-off of ≥8 on the depression domain indicates depression among a general population (19).

Health-related quality of life was assessed using the EuroQoL-5D-5L (EQ-5D-5L) and its VAS (22). The EQ-5D-5L assesses five domains (mobility problems, self-care problems, usual activities problems, pain/discomfort, and anxiety/depression) with 5 response options ranging from “no problems” to “extreme problems or unable to” (23). The VAS characterizes overall health with scores ranging from 0 (worst imaginable health state) to 100 (best imaginable health state) (22). The EQ-5D-5L has excellent measurement properties across a broad range of ill populations (24).“

R2: Variables

Clearly define all outcomes, exposures, predictors, potential confounders, and effect modifiers.

Response: We agree that it is important to describe the details of data sources and measurement approaches. We have extensively described the outcome measurement tools we have used across the study on lines 126-185.

R2: Give diagnostic criteria

Response: Thank you for this comment. We have expanded and clarified the diagnosis criteria for patient eligibility on lines 106-111 as follows:

“We enrolled adult patients (≥18 years) who were either currently or recently hospitalized for a COVID-19 infection, as defined by the International Severe Acute Respiratory and Emerging Infection Consortium. Using daily Infection Prevention and Control data, site leads identified potential patients admitted to a medical unit, emergency department or intensive care unit based on either a confirmed positive nasopharyngeal swab or a documented COVID-19 diagnosis. “

R2:

Bias

Describe any efforts to address potential sources of bias

Response: Thank you for this comment. We have addressed potential sources of bias on lines 321-329 where we describe the limitations of the study. Specifically, we acknowledge that reliance on self-reported data—although collected using validated questionnaires—may introduce recall or reporting bias. Additionally, there is a potential for selection bias, as participants who agreed to complete the 24-month follow-up may differ systematically from those who declined. However, with the exception of post-secondary education, no consent differences were observed by sex, age, household income, hospital length of stay, or baseline AM-PAC mobility or cog

---

## [Editor Report · Decision Letter 1]

25 Sep 2025

Functional recovery 2-years after hospitalization for COVID-19: Insights from the COREG-FR extension study

PONE-D-25-16167R1

Dear Dr. Beauchamp,

We’re pleased to inform you that your manuscript has been judged scientifically suitable for publication and will be formally accepted for publication once it meets all outstanding technical requirements.

Kind regards,

Armaan Jamal

Guest Editor

PLOS ONE

---

## [Editor Report · Acceptance letter]

PONE-D-25-16167R1

PLOS ONE

Dear Dr. Beauchamp,

I'm pleased to inform you that your manuscript has been deemed suitable for publication in PLOS ONE. Congratulations! Your manuscript is now being handed over to our production team.

Kind regards,

on behalf of

Mr. Armaan Jamal

Guest Editor

PLOS ONE